# Precision Oncology: Evolving Clinical Trials across Tumor Types

**DOI:** 10.3390/cancers15071967

**Published:** 2023-03-25

**Authors:** I-Wen Song, Henry Hiep Vo, Ying-Shiuan Chen, Mehmet A. Baysal, Michael Kahle, Amber Johnson, Apostolia M. Tsimberidou

**Affiliations:** 1Department of Investigational Cancer Therapeutics, The University of Texas MD Anderson Cancer Center, 1515 Holcombe Blvd., Houston, TX 77030, USA; 2Khalifa Institute for Personalized Cancer Therapy, The University of Texas MD Anderson Cancer Center, 1515 Holcombe Blvd., Houston, TX 77030, USA

**Keywords:** precision oncology, investigational therapy, clinical trials, targeted therapy, immunotherapy, IMPACT, TAPUR, NCI-MATCH

## Abstract

**Simple Summary:**

Precision oncology is the use of anticancer drugs to specifically inhibit the function of aberrant oncogenic proteins driving a patient’s tumor. The application of molecular technologies and targeted therapeutics has led to significant advancements in precision oncology, resulting in favorable clinical outcomes for selected patients with cancer. This review focuses on selected precision oncology clinical trials that match patient- and tumor-specific aberrations with targeted therapies. These trials include the IMPACT, SHIVA, IMPACT2, NCI-MPACT, TAPUR, DRUP, and NCI-MATCH trials. Significant and impactful progress has been made towards the realization of precision oncology, and many matched targeted therapies are now available for patients with cancer. However, precision oncology remains inaccessible to many patients. The successes, challenges, and opportunities that have emerged—and the lessons learned—are highlighted. The use of artificial intelligence, machine learning, and bioinformatic analyses of complicated multi-omic data may improve the tumor characterization process and accelerate the implementation of precision oncology.

**Abstract:**

Advances in molecular technologies and targeted therapeutics have accelerated the implementation of precision oncology, resulting in improved clinical outcomes in selected patients. The use of next-generation sequencing and assessments of immune and other biomarkers helps optimize patient treatment selection. In this review, selected precision oncology trials including the IMPACT, SHIVA, IMPACT2, NCI-MPACT, TAPUR, DRUP, and NCI-MATCH studies are summarized, and their challenges and opportunities are discussed. Brief summaries of the new ComboMATCH, MyeloMATCH, and iMATCH studies, which follow the example of NCI-MATCH, are also included. Despite the progress made, precision oncology is inaccessible to many patients with cancer. Some patients’ tumors may not respond to these treatments, owing to the complexity of carcinogenesis, the use of ineffective therapies, or unknown mechanisms of tumor resistance to treatment. The implementation of artificial intelligence, machine learning, and bioinformatic analyses of complex multi-omic data may improve the accuracy of tumor characterization, and if used strategically with caution, may accelerate the implementation of precision medicine. Clinical trials in precision oncology continue to evolve, improving outcomes and expediting the identification of curative strategies for patients with cancer. Despite the existing challenges, significant progress has been made in the past twenty years, demonstrating the benefit of precision oncology in many patients with advanced cancer.

## 1. Introduction

Precision oncology is the use of anticancer drugs to inhibit the function(s) of biological or molecular alterations identified in an individual patient’s tumors or circulating tumor DNA. The practice of precision oncology began with the discovery of imatinib for the treatment of newly diagnosed Philadelphia-chromosome-positive chronic myeloid leukemia [1] and has expanded to include the development of novel therapeutic agents that target biological abnormalities associated with cancer growth, and most recently immunotherapy [2,3,4,5]. Advancements in and increased access to genome sequencing technology, the accumulation of knowledge from basic research, and the translation of basic research findings into clinical trials to develop effective anticancer therapies have all led to an increase in the use of precision oncology for the therapeutic management of patients with cancer. Prior to the use of precision oncology, the treatment was selected on the basis of the anatomical origin of the tumor, without taking into consideration the tumor biology of individual patients. Since the precision oncology approach was initiated, multiple clinical trials have been conducted. In this review, published data from selected completed or ongoing trials in precision oncology that provide clinical outcomes are summarized, including the MD Anderson IMPACT studies; the randomized SHIVA study; and the NCI-MPACT, TAPUR, DRUP, and NCI-MATCH trials.

## 2. Biomarker-Selected Clinical Trials

Biomarkers are genomic alterations or molecular profile signatures of an individual’s tumor that may have diagnostic, prognostic, or therapeutic implications. To accrue patients on a specific biomarker-selected precision oncology trial, various requirements should be met. Potential participants must not only have the specified biomarker and tumor type (as well as meet the other eligibility criteria of the study) but must also be identified at the right time in their treatment course. In addition, the clinical trial should be immediately available for enrollment.

Accurately determining a biomarker match on the basis of a patient’s molecular profile is a complex process that depends on the background and training in genomic testing of the investigators performing the matching (e.g., the oncologist, decision support scientist, or other health care provider). Although the patient selection is straightforward if a clinical trial is enrolling patients with a specific, clearly defined alteration, many cases are complicated. An overview of molecular profiling in precision oncology is depicted in Figure 1.

### 2.1. Biomarker Nomenclature, Hierarchy, and Reporting Format

A biomarker required for enrollment on a clinical trial may be described differently in the eligibility criteria of the trial than on a patient’s sequencing report. Thus, the treating physician must determine if the biomarker is indeed a match [6]. First, regarding the gene-level nomenclature, investigators should be aware of gene aliases. For example, if the trial specifies selection for alterations in *HER2*, *MLL*, or *LKB1*, genomic alterations in *ERBB2*, *KMT2A*, or *STK11*, respectively, in the patient sequencing reports should be recognized as being in the same genes. Second, trials often do not select for specific genomic variants but rather a class or type of alteration, and the treating physician must decide if the patient’s detected variant meets the requirements set within the trial eligibility criteria. For example, some trial documents may use the term “BRCA-positive” tumors while recruiting patients with deleterious or suspected deleterious BRCA1 or BRCA2 loss-of-function mutations or deletions. Additionally, if a trial selects for a particular mutation subtype, such as EGFR exon 19 deletions, the investigators should be familiar enough with the nomenclature to identify the alterations that are small in-frame deletions and that also occur within the amino acid range of EGFR exon 19. Indeed, familiarity with the gene nomenclature is also necessary for investigators to distinguish alteration types and the alteration itself. Third, specific next-generation sequencing (NGS) tests may not sequence all exons in the genes that the panel is reported to cover and may only sequence or report alterations at hotspots or known actionable regions. Additionally, an NGS test may sequence and report alterations of one alteration type, e.g., mutations, but not others, e.g., fusions or copy number alterations. Thus, if an alteration is not reported on a hotspot panel or if a panel with restricted coverage, including of alteration types, is used, this does not necessarily mean that no alterations are present, which is essential if a trial requires the wild-type status of a gene. However, major NGS tests have expanded their coverage and transparency. Fifth, when matching a particular patient’s tumor molecular alteration to an accruing trial, the investigators must be aware of the somatic, germline, or indistinguishable status of the patient’s reported alteration and specific genomic markers being selected for the trial. A timeline of selected clinical trials across tumor types in precision oncology is depicted in Figure 2.

### 2.2. Biomarker and Literature Evolution

The molecular evolution of a patient’s tumor and the evolving data in the field of precision oncology should also be considered in biomarker matching. As NGS has become a routine test, investigators should consider whether a sequencing report from several years ago accurately reflects the current molecular profile of a patient’s tumor. Furthermore, if the test was performed on a previous tumor or a tumor from a separate disease site, it may not fulfill the current clinical need. Likewise, the body of literature in precision oncology has drastically expanded over the past several years. An alteration that may have been reported in a “variants of uncertain or unknown significance” section several years ago could now be well established in the literature to have functional significance or therapeutic implications [7,8].

## 3. Clinical Trials in Precision Oncology across Tumor Types

### 3.1. Initiative for Molecular Profiling and Advanced Cancer Therapy (IMPACT)

Despite the prevailing notion that tumor molecular profiling for solid tumors would not be efficient for selecting treatment, in 2007 we initiated the IMPACT (Initiative for Molecular Profiling and Advanced Cancer Therapy) precision oncology study. IMPACT was an exploratory, non-randomized study that enrolled patients with advanced solid tumors who were referred to the Phase I Clinic at The University of Texas MD Anderson Cancer Center (Houston, TX) for investigational therapy (NCT00851032).

The hypothesis was that the evaluation of molecular profiling and selection of molecularly driven therapy would be associated with favorable outcomes in patients with advanced cancer. Sequential patients with advanced metastatic cancer who were considered for investigational therapy and underwent tumor molecular profiling were included in the analysis. The allocation of patients to investigational treatments varied over time according to the protocol availability, eligibility criteria, histologic diagnosis, patient’s prior response to therapy, potential toxicity, insurance coverage, and patient preference or physician choice. Physicians prioritized matched therapy (vs. non-matched therapy) on the basis of the presence of an “actionable” molecular aberration and the availability of matched targeted therapy. Patients with “actionable” molecular aberrations who met the study criteria were treated with matched targeted therapy, when available [9].

In the first analysis (2011), 40.2% of 1144 patients had at least one genomic aberration. Among the genes and amino acid ranges covered by assays in this analysis, the most common alterations found were TP53 mutations, KRAS mutations, PTEN loss, BRAF mutations, and PIK3CA mutations across solid tumors, as well as RET mutations in medullary thyroid cancer specifically. The most common tumor types were melanoma, thyroid, colorectal, endometrial, lung, pancreatic, and breast cancers.

Overall, 175 patients were treated with matched targeted therapy (MTT), and 116 with non-matched therapy (non-MTT). The treatments pursued predominantly consisted of small-molecule kinase inhibitors, such as PAM (PI3K/AKT/mTOR) pathway inhibitors, RAF/MET inhibitors, KIT, EGFR, and RET. Many of the targeted agents had multikinase inhibitory activity. The overall response rates (ORRs) were 27% and 5% for matched and non-matched cohorts, respectively (*p* < 0.0001). The time to treatment failure (TTF) and overall survival (OS) were also longer in the MTT group compared to the non-MTT group (Table 1). Patients treated with matched targeted therapy also had a longer time to treatment failure when compared with their prior systemic therapy (5.2 vs. 3.1 months; *p* < 0.0001) [9]. This trial influenced future trials in precision oncology, leading to multiple trials across tumor types.

These data were validated in a second cohort treated with a similar approach [11]. In a multivariate analysis, MTT was an independent factor predicting the response and PFS. Two-month landmark analyses in the MTT group demonstrated that the median OS of the responders was 30.5 months compared with 11.3 months for the non-responders (*p* = 0.01) [11]. In a third patient cohort [12], 637 of 1436 patients had at least one actionable aberration. MTT was associated with higher rates of ORR, FFS, and OS. Interestingly, patients with phosphoinositide 3-kinase (PI3K) and mitogen-activated protein kinase (MAPK) pathway alterations matched to PI3K/AKT serine–threonine kinase/mammalian target of rapamycin (mTOR) axis inhibitors alone demonstrated outcomes comparable to unmatched patients [12]. In a long-term analysis of 3487 patients who completed tumor molecular testing from September 2007 to December 2013, 711 received MTT and 596 received non-MTT [13]. The 10-year OS rates were 6% versus 1%, respectively, for the MTT and non-MTT groups (HR = 0.72; *p* < 0.001), and MTT was an independent factor predicting longer OS [13]. Thus, our collective experience with the precision oncology approach was encouraging (Table 1).

These analyses demonstrated in independent patient sets that molecularly based matched targeted therapy was associated with superior rates of response, PFS, and OS compared to non-targeted therapy. Furthermore, these results stimulated the development of efficient pipelines to allow timely molecular testing, interpret patient molecular profiles, design biomarker-matched trials, and accelerate drug development. Since we started the IMPACT study in 2007, the number of validated genes that could be routinely molecularly analyzed for patient care and the number of approved targeted therapies and clinical trials have significantly increased.

However, IMPACT had many limitations. In many cases, archival tissue was used. Additionally, some patients were treated on clinical trials with matched targeted agents combined with a cytotoxic agent; therefore, the results may be attributed to synergistic effects. The benefits of matched targeted therapy may be diminished when low doses of targeted agents that have been ultimately proven to perform poorly in the human setting are used. The complexity of tumor biology in the advanced setting may limit the antitumor activity of matched targeted therapy against single molecular alterations. More importantly, the results were not derived from a randomized trial, and unknown confounding factors may have contributed to the superior outcomes noted with matched targeted therapy. To overcome the challenges noted with IMPACT, we designed IMPACT2, a randomized study in precision oncology.

### 3.2. SHIVA, a Study of Randomized, Molecularly Targeted Therapy Based on Tumor Molecular Profiling versus Conventional Therapy for Advanced Cancer

SHIVA (NCT01771458) was the first randomized, controlled phase 2 trial in precision oncology across tumor types. The aim was to assess whether the use of molecularly targeted agents outside their indications could improve patient outcomes if given according to a predefined treatment algorithm and based on the molecular alterations identified [18]. The investigators included patients who had alterations in hormone receptors, PI3K/AKT/mTOR, or RAF/MEK pathways. Randomization was achieved centrally using a web-based response system based on the Royal Marsden Hospital prognostic score (0 or 1 vs. 2 or 3) and the altered molecular pathway. Overall, 741 patients were enrolled (October 2012–July 2014), and the primary endpoint was PFS. The investigators reported no difference in PFS between patients treated with MTT and those treated according to physician choice (HR: 0.88; 95% CI, 0.65–1.19; *p* = 0.41) [18]. The study was conducted in multiple institutions in France, highlighting that significant resources are required to conduct a randomized study in precision oncology. It also demonstrated that a multidisciplinary team that accurately executes each step in the process—patient enrollment, tumor biopsy, tissue processing, NGS, alteration annotation, treatment matching, and extensive safety monitoring for patients on treatment—is essential for the success of precision oncology. However, no difference in outcomes was demonstrated between the two arms, likely owing to limitations in the study design [18,19]. For instance, multiple molecular alterations are unlikely to respond to monotherapy [20], and 80% of the patients received everolimus or hormone blocker monotherapy. Everolimus is ineffective even in the presence of a match [12] if multiple genomic alterations are present, and it is unlikely for previously treated patients who have hormone receptor abnormalities to respond to hormonal monotherapy. In addition, imatinib, an ineffective RET inhibitor, was matched to RET alterations. Finally, bias could have been introduced into the study because a predefined algorithm was used to assign targeted therapies, whereas physicians assigned the therapies to the control group [19].

### 3.3. Initiative for Molecular Profiling and Advanced Cancer Therapy II (IMPACT2)

To determine whether tumor molecular profiling to select treatment is superior to treatment selection not based on molecular profiling, in 2014 we initiated IMPACT2, a large randomized controlled trial (RCT) in precision oncology, as RCTs are considered the gold standard for the evaluation of the cause-and-effect relationship of an intervention and an endpoint [21,22]. IMPACT2 (NCT02152254), an ongoing phase 2 RCT (randomization rate, 1:1) with an adaptive, innovative study design, focuses on the use of molecular testing and targeted therapy across tumor types. The endpoint of IMPACT2 is PFS in the MTT versus non-MTT patient groups. IMPACT2, like other RCTs, is arduous. Originally (part A), patients who met the criteria for randomization were randomized between two arms (MTT vs. non-MTT). However, evolving data in precision oncology and increasing interest in incorporating patient preference into the treatment selection led to the trial being amended in March 2019 to include a “patient preference” cohort (part B). According to the revised design, patients eligible for randomization can now select their preferred arm or choose to be randomized between the two arms. Notably, both arms include investigational therapy, and patients provide informed consent (in addition to consent to participate in IMPACT2) stating that they are aware of the investigational nature of the individual clinical trials or treatments.

As of October 2021, 600 patients (part A, n = 391; part B, n = 209) were enrolled in the study, and 85 had been randomized. Of 474 patients with at least one targetable alteration, 230 (48.5%) patients had tumor protein P53 (*TP53*) alterations. Other commonly detected molecular alterations included cell-cycle-associated genes (34.8%), PI3K/AKT/mTOR pathway alterations (30.8%), and MAPK signaling abnormalities (28.4%). In part A, 326 patients completed molecular testing and 317 (97.2%) patients had at least one aberration (targetable, n = 191; non-targetable, n = 126). Overall, 21.1% of the patients were randomized. Of the remaining patients who were not randomized, 61% were treated with an investigational or standard therapy. In part B, 91.3% of 162 patients who completed the tumor molecular profiling had at least one targetable alteration. We offered randomization to 32 patients who met the criteria to be randomized, and 50% accepted to be randomized. The remaining patients selected their treatment arm. The outcomes will be reported at the time of study completion.

As with other precision oncology or randomized studies, we experienced the following challenges: (1) on average, the time from patient enrollment to biopsy was seven days and the time from biopsy to the availability of molecular profiling results was 19 days, most patients required immediate therapy, and upon progression they were ineligible to act on the molecular profiling and participate in clinical trials owing to their deteriorating performance status and organ function [23]; (2) many patients lacked actionable tumor molecular alterations; (3) patients were ineligible for clinical trials owing to comorbidities or clinical trials were unavailable; (4) patients did not have the resources to comply with treatment protocol requirements; (5) some participants in the randomized arm of IMPACT2 did not get the assigned therapy because their insurance did not cover the cost [23].

There are still many barriers to overcome. In addition to the complex biology and plasticity of tumors, the numerous molecular alterations that occur in the advanced metastatic setting in which clinical trials are typically developed cannot be addressed by the currently available drugs, apart from checkpoint inhibitors, which are used to treat tumors that exhibit a high tumor molecular burden [23]. Additionally, the identified molecular alterations might not correspond to the causative biomarker(s), or the molecular environment may be distinct between the primary tumor and the metastatic areas [24]. Ct-DNA analysis may shorten the time it takes to acquire molecular profiling results, limiting the need for bridging therapy, and may overcome the differences between primary tumor and metastatic sites [25,26,27]. Importantly, single-agent treatment modalities likely offer only temporary improvements; therefore, innovative drug combinations and strategies should be developed.

This ongoing single-institution trial in precision oncology indicates the challenges associated with and the multiple resources that are required to conduct a randomized trial in precision oncology. Timely patient enrollment, fresh tumor biopsies, molecular tumor board reviews, the availability of targeted agents (off-label or through clinical trials), the timely initiation of treatment, the assessment of the response and toxicity, and the close monitoring of patients are required.

### 3.4. National Cancer Institute Molecular Profiling-Based Assignment of Cancer Therapy (NCI-MPACT)

NCI-MPACT (NCT01827384), a phase II RCT initiated in 2013, used tumor DNA sequencing for treatment selection in patients with advanced cancer and somatic mutations to the DNA repair pathway, the RAS/RAF/MEK pathway, or the PI3K/Akt/mTOR pathway [28]. The primary endpoint was the ORR with the first regimen used. The patients were randomized (2:1) to receive either a study regimen identified to target the aberrant pathway found in their tumor or one of the remaining three regimens not targeting that pathway. Of 49 randomized patients, one (5%) of 20 patients in the experimental trametinib cohort had a PR [28]. This study demonstrated a very low rate of objective response, indicating the need for highly effective, tailored therapy targeting specific genetic aberrations for the implementation of precision oncology. It was challenging to randomly assign patients to a non-targeted control arm. Some patients and physicians possibly had prior tumor mutation profile knowledge and declined to participate in the control arm [28].

### 3.5. The National Cancer Institute’s Molecular Analysis for Therapy Choice (NCI-MATCH)

NCI-MATCH (NCT02465060) was launched in 2015, enrolled patients with advanced refractory solid tumors, myeloma, or lymphoma, and was designed to evaluate MTTs based on specific, actionable molecular tumor alterations [29]. The primary endpoint of the study was to evaluate the objective response rates in patients with advanced refractory cancers treated with matched targeted therapies. The secondary endpoints were to evaluate the rates of PFS and OS at 6 months and to identify potential predictive biomarkers beyond the genomic alteration by which the treatment was assigned; to identify resistance mechanisms using additional genomic ribonucleic acid (RNA), protein, and imaging-based assessment platforms; and to assess whether radiomic phenotypes obtained from pre-treatment imaging and changes from pre- through post-therapy imaging can predict the objective response and progression-free survival and their association with targeted gene mutation patterns of tumor biopsy specimens.

The patients underwent a tumor biopsy after study enrollment or archival tissue samples collected within the previous six months were used for the assessments, which included single-nucleotide variants, indels, amplifications, and selected fusions using a 143-gene NGS panel and PTEN, MLH1, and MSH2 expression using immunohistochemistry [30,31]. The actionability of the molecular alterations was determined based on the availability of FDA-approved drugs targeting the alteration, ongoing trials accruing for the specific alterations, or available robust preclinical data. A prospectively defined NCI-designed informatics rules algorithm (MATCHBOX) was used to assign patients to one of 39 treatment arms (subprotocols), which involved individual eligibility screening [32]. The rates of participant enrollment [33,34], the results from some of the subprotocols [35,36,37,38,39,40], and the investigators’ reports have shown the feasibility of the molecular profiling and treatment assignment processes [33]. As of August 2022, 1199 patients had been accrued across all treatment arms [29].

Positive results have been reported for five of the arms [35,41,42,43,44]. In patients with cancers other than colorectal cancer with mismatch repair deficiency, treatment with nivolumab was associated with a 12-month PFS rate of 46.2% and a median OS duration of 17.3 months [35]. In the cohort of patients with AKT1 E17K mutations treated with capivasertib, the ORR was 28.6%, the 6-month PFS rate was 50%, and the median OS duration was 14.5 months [41]. The primary endpoint was positive for the dabrafenib and trametinib treatment cohort of patients with BRAF V600E mutations, with an ORR of 38% [42]. The cohort of copanlisib in patients whose tumors harbored PIK3CA mutations also met its primary endpoint (ORR, 16%) [43]. In patients with tumors harboring ALK or ROS1 rearrangements, although the accrual number was low (n = 4), the responses to crizotinib treatment met the primary endpoint for the ALK fusion group [45]. Negative results have been reported for six cohorts [37,46,47,48,49,50], and the remaining subprotocols are ongoing [29] (Table 2 and Table 3). Two subprotocols are open for enrollment (arm H, expansion phase, dabrafenib and trametinib targeting BRAF V600E or V600K mutations; arm Z1M, relatlimab and nivolumab targeting LAG-3 expression with microsatellite instability), and 14 arms are closed [29].

The advantages of NCI-MATCH included providing access to molecular profiling (first part of the study) and matched targeted therapy to many patients at participating academic and community centers in the U.S. [33]. The challenges included a delay in the molecular profiling results and the very small proportion of patients who received matched targeted therapy in the first part of the study owing largely to a lack of trial availability. Many screened patients were ineligible for drugs in the initial 10 subprotocols because they had tumor types for which the initial treatments were FDA-approved. In addition, the time period patients were required to be off treatment (4–6 weeks) for molecular profiling and inadequate laboratory resources to provide molecular profiling in a timely manner contributed to a worsening patient performance status and a low enrollment rate (48.5%).

A lesson learned was the significance of an interim analysis in such large studies. After this analysis, 24 subprotocols were available, and the assignment rate increased from 5.1% to 25.3%. There was a great demand for molecular profiling studies, indicating the need for adequate laboratory resources to provide results reliably and in a timely manner. The successes included the training and engagement of participating health care organizations about the specifics of tissue molecular profiling and the initiation of targeted therapy as well as the NCI’s National Clinical Trials Network (NCTN) and National Community Oncology Research Program, which included investigators familiar with new drug administration and access to many patients with malignancies. This collaboration, led by the NCI and ECOG-ACRIN, with participation from representatives from all the NCTN groups and the incorporation of expert input from study committees, principal investigators, and the pharmaceutical industry, was essential for success.

The implementation of the approach was uniform under an NCI-sponsored Investigational New Drug Application, as well as by use of the NCI Central Institutional Review Board. In the phase II cohorts of uncommon alterations (prevalence range, 1–2%), patients with molecular profiling in a CLIA-certified laboratory were enrolled in the study to receive targeted therapy [33].

### 3.6. Targeted Agent and Profiling Utilization Registry (TAPUR)

TAPUR is a phase II, prospective, non-randomized, open-label basket study that evaluates the antitumor activity of commercially available targeted agents in patients whose advanced cancers have genomic alterations that are targets for these drugs. This precision oncology trial was designed and led by the American Society of Clinical Oncology, matching patients’ tumor genomic alterations identified in a CLIA-certified laboratory to off-label, FDA-approved, targeted anticancer agents (NCT02693535) [72]. The primary endpoint of the study is the disease control rate, defined as the CR or PR at 8 weeks or later or stable disease at 16 weeks or later from the initiation of study treatment. The secondary endpoints include the PFS, OS, and safety. TAPUR was initiated in March 2016, and as of August 2022, positive results had been reported for 15 cohorts [51,52,54,55,56,57,58,59,60,61,62,63,64] (Table 2).

In patients with BRCA1/2-inactivating mutations treated with olaparib, those with prostate cancer had a median PFS duration of 41.0 weeks and a median OS duration of 75.4 weeks (1-year OS rate, 79.4%) [51], while those with pancreatic cancer had a median PFS duration of 8.1 weeks and a median OS duration of 43.0 weeks (1-year OS rate, 42.7%) [52]. In patients with diverse cancers, the median PFS duration was 15.7 weeks and the median OS duration was 45.0 weeks [54]. Olaparib was also associated with antitumor activity in patients with a mutation or deletion of ATM (median PFS duration of 8.6 weeks; median OS duration of 40.9 weeks) [53].

Positive results were also reported with palbociclib. In patients with head and neck cancer bearing a CDKN2A loss or mutation treated with palbociclib, the median PFS duration was 9.4 weeks and the median OS duration was 42.0 weeks [55]. In patients with non-small cell lung cancer and tumoral CDKN2A alterations, the median PFS duration was 8.1 weeks and the median OS duration was 21.6 weeks [56]. Finally, for patients with CDK4-amplified soft tissue sarcoma, the use of palbociclib was associated with a median PFS duration of 16.1 weeks and a median OS duration of 68.7 weeks (1-year OS rate, 53.6%) [57].

In the pertuzumab plus trastuzumab cohort for patients whose tumors had ERBB2 (or ERBB3) alterations, those with colorectal cancer had a median PFS duration of 17.2 weeks (1-year OS, 58%) [58]; those with uterine cancer had a median OS duration of 28.1 weeks (1-year OS rate, 53.4%) [59]; and those with lung cancer had a disease control rate (DCR; overall response or SD ≥ 16 weeks) of 37% with a median OS duration of 54.4 weeks [60].

Among the patients with a high tumor mutational burden (TMB) treated with pembrolizumab, in the patients with metastatic breast cancer, the DCR was 37%, and the durations of PFS and OS were 10.6 weeks and 30.6 weeks, respectively [61]. In patients with CRC, the DCR was 28%, the median duration of PFS was 9.3 weeks, and the 1-year OS rate was 45.6% [62]. Notably, pembrolizumab was approved by the FDA in June 2020 for the treatment of patients with unresectable or metastatic solid tumors that were TMB-high (≥10 mutations/megabase) without alternative treatment options.

In addition, the use of sunitinib in patients with heavily pre-treated metastatic breast cancer and tumor FGFR1 amplification was associated with a DCR of 29%, a median PFS duration of 8.7 weeks, and a median OS duration of 33.9 weeks [63].

Positive results were also reported for the cobimetinib plus vemurafenib treatment cohorts targeting BRAF mutations, including colorectal cancer with BRAF V600E mutations, with a DCR of 57%, median PFS duration of 15.8 weeks, and median OS duration of 38.9 weeks [64], and diverse tumors with BRAF V600E/D/K/R mutations, with a DCR of 68%, a median PFS duration of 5.8 months, and median OS duration of 15.2 months [65].

While negative results were reported for seven cohorts with the targeted agents [66,67,68,69,70,71] (Table 3), the remaining 16 cohorts were still ongoing at the time of this review, and the findings were pending [73].

Similar to the NCI-MATCH study, TAPUR provides access to matched targeted therapies to patients across tumor types in many institutions and practices in the U.S. The patients’ molecular profiling is being reviewed by a molecular tumor board when their molecular alterations are not a clear match with the available treatment cohorts. The study drugs are provided to ASCO by pharmaceutical companies and then they are provided to the patients at no cost. The clinical outcomes are recorded by the treating physicians and are reported by tumor type and molecular alterations.

### 3.7. The Drug Rediscovery Protocol (DRUP) Trial

DRUP is a prospective, non-randomized clinical trial that aims to describe the efficacy and toxicity of commercially available, targeted anticancer agents prescribed for the treatment of patients with advanced cancer with a potentially actionable genomic or protein expression variant. The study design shares some similarities with TAPUR but was conducted as an independent protocol in the Netherlands. The expected outcome is that using approved drugs in new ways based on the molecular profiling derived from fresh biopsies of patient tumors leads to better treatment options and results and greater access to targeted therapy (NCT02925234). The goal is to improve and expand the use of registered targeted therapy, while making it more accessible to patients who have exhausted standard treatment options. A tumor board helps physicians understand the profiling test results and treatment options.

The investigators recently reported the potential benefits of using off-label matched targeted therapy in patients with treatment-refractory metastatic cancers and rare cancers (incidence, <6 cases per 100,000 persons per year) [74]. Patients harboring an actionable molecular alteration were matched with off-label targeted therapies or immunotherapies that were approved by the FDA or European Medicines Agency. The patients were enrolled in separate cohorts based on their histologic tumor type, molecular profile, and the study drug they received. The primary endpoint of the study was to determine the clinical benefit, which includes CR, PR, and SD lasting ≥ 16 weeks. In an analysis of 1145 patients with cancer, 500 patients (including 164 patients with rare cancers) initiated therapy with one of 25 drugs (off-label) and were evaluable for outcomes. Overall, 33% of patients with both rare and non-rare cancers experienced clinical benefits. Patients with rare cancers more frequently had CDKN2A and BRAF genetic alterations compared to patients with non-rare cancers, leading to more matches with CDK4/6 inhibitors (14% vs. 4%; *p* ≤ 0.001) or BRAF inhibitors (9% vs. 1%; *p* ≤ 0.001). Patients with rare cancers treated with off-label BRAF inhibitors had a 75% clinical benefit rate, higher than the non-rare cancer group.

The investigators demonstrated that molecular testing can help identify beneficial treatment options for patients with rare cancers as well as those with common cancers, providing access to broad molecular diagnostics and equal treatment opportunities for all patients with cancer [74].

### 3.8. Other Clinical Trials Focusing on Advanced Diverse Cancers

Other important clinical trials in precision oncology across tumor types have shown encouraging results. The I-PREDICT (Investigation of Profile-Related Evidence Determining Individualized Cancer Therapy) study is a prospective navigation trial for patients with refractory or therapy-naive metastatic cancers [14]. The tumor genomic profiling, ctDNA analysis, programmed death-ligand 1 (PD-L1) expression, hormonal status, tumor mutational burden, and microsatellite instability status were evaluated and scored to inform the multidrug combination selection proposed by the multidisciplinary molecular tumor board. The clinical benefit (stable disease ≥ 6 months/partial response/complete response) rate with matched vs. unmatched therapy, respectively, was 34.5% vs. 16.1% (*p* ≤ 0.005; *p* = 0.02 multivariable or propensity score methods), and the median PFS was 4.0 vs. 3.0 months (*p* = 0.039 in the Cox regression model) [14]. The investigators calculated the matching score, defined as the number of matched drugs divided by the number of aberrations. The unmatched patients had a score of 0. The median OS of the patients with a matching score > 0.2 was 15.7 months compared with 10.6 months for the patients with a matching score ≤ 0.2, (*p* = 0.040, Cox regression model) [14].

WINTHER, a clinical trial across tumor types conducted by the Worldwide Innovative Network (WIN) Consortium, navigated personalized cancer therapy using genomics (arm A) or transcriptomics (arm B) (NCT01856296) [15]. Based on the IMPACT study [9] and the Von Hoff model [10], a PFS improvement ratio > 1.5 was observed in 18/66 patients, with a higher degree of matching with molecular supporting information, either genomic or transcriptomic, than with the prior treatment selection. This study demonstrated that genomic and transcriptomic profiling are helpful for improving therapy recommendations and patient outcomes [15].

The DART (Dual Anti-CTLA-4 and Anti-PD-1 blockade in Rare Tumors) study (NCT01856296) focused on 50 rare cancer histological types and immunotherapy interventions. This study demonstrated the clinical benefit of ipilimumab plus nivolumab in high-grade non-pancreatic neuroendocrine neoplasms with an ORR of 44% (8/18 patients) [16] and in unresectable metastatic breast cancer with an ORR of 18% (3/17 patients) [75].

A phase I/II Octopus study, Quilt-3.055 (NCT03228667), recruited patients with diverse cancer types and focused on combining T cell modulation and PD-L1 inhibition for advanced cancer patients previously treated with immunotherapy [17]. The preliminary partial response rate was 8%, the stable disease rate was 51%, the median PFS was 3.9 months, and the median OS was 13.8 months [17].

## 4. Future Trials

The clinical trials in precision oncology continue to expand. For example, the NCI is launching new studies that include ComboMATCH, MyeloMATCH, and iMATCH. The ComboMATCH study is a phase II trial that focuses on the investigation of targeted drug combinations, based on the gene signatures of specific cancers, in order to overcome drug resistance to single-agent therapy [76]. The study aims to identify genetic mutations linked with the responses to targeted therapy combinations and has the potential to generate more individualized and efficacious treatments for patients with various solid tumors, including lung, breast, colon, and pancreatic cancers. The primary objective of ComboMATCH is to overcome the drug resistance to single-agent therapy and to enhance the effectiveness by developing genomically directed combination therapies. These therapies are designed to leverage new synergies that are supported by compelling evidence from preclinical in vivo studies. ComboMATCH employs single-arm and randomized designs. Unlike the NCI-MATCH trial, it will incorporate children in the same trial rather than having a separate pediatric MATCH trial in parallel [76].

The aim of MyeloMATCH is to expedite drug development for patients with newly diagnosed acute myeloid leukemia (AML) and myelodysplastic syndromes (MDS) [77], establishing many rationally designed substudies. The study includes 4 tiers and 5 clinical baskets. Tier 1 includes phase 2 randomized studies for initial therapies grouped by disease (MDS, younger AML, and older AML), using novel drug combinations with a measurable residual disease (MRD) assessment conducted centrally. The subsequent therapy occurs in higher tiers, and assignments are made on the basis of prior treatment substudy outcomes. Flow cytometry and sequencing will be employed in tier 4 clinical trials that focus on residual disease. The clinical utility of the assays and biomarkers to determine if targeting residual disease confers clinical benefit will also be assessed.

In summary, as patients progress to higher tiers with a lower remaining tumor burden, the primary focus will shift towards the more precise targeting of residual disease. The early endpoints include the identification of significant activity signals that could generate promising data for further definitive studies and optimizing the use of resources for generating reliable and high-quality randomized trial data that can aid in the selection of phase 3 priorities. The new collaborative model for conducting clinical trials may lead to significant breakthroughs for patients with AML and MDS [77].

The iMATCH study will focus on the evaluation of patients’ immunologic profiles and immune markers and the selection of specific trials [78]. The primary endpoint of the study is to assess the objective response rate within and across four biomarker subgroups categorized on the basis of their high or low TMB and tumor inflammation score, which will be used for the identification of substudies for patient enrollment. The pilot study, S2101, features a combination cabozantinib and nivolumab treatment for patients with locally advanced or metastatic melanoma or head and neck cancer whose disease has progressed while on previous immunotherapy [78].

## 5. Conclusions and Future Directions

Precision oncology has entered a new era owing to advancements in technology, a deeper understanding of the mechanisms of carcinogenesis, and the development of new, more effective anticancer agents. Many trials in precision oncology have similar workflows that include NGS and other biomarker analyses of tumor or blood samples, the interpretation of the results, and treatment with matched targeted therapy. However, some differences do exist. For instance, some investigators use in-house CLIA-validated sequencing panels, whereas others use commercially available laboratories. The gene panels and the inclusion of immune biomarkers such as PDL1 testing, the tumor mutational burden, and the microsatellite instability status may also differ between clinical trials and centers. Within those pipelines, the bioinformatics methods for identifying and reporting alterations may also vary. In recent years, the panels have expanded from dozens of genes to a few hundred genes using NGS and to a few thousand genes using whole-exome sequencing. Additionally, some investigators have established molecular tumor boards with dedicated decision support experts for the interpretation of genomic and molecular abnormalities and for matching alterations with indications following established guidelines or on the basis of selection criteria for clinical trials. However, these molecular and biomarker tests (other than those certified by CLIA, the AMP (Association for Molecular Pathology), or the ISO (International Organization for Standardization)) are not standardized, and the molecular tumor boards and access to drugs may vary. The standardization of testing, access to targeted therapies, and continued harmonization between translational precision oncology policies and practices are required to implement precision oncology.

In the last fifteen years, significant progress has been made and many drugs are now available on the basis of molecular profiling. As of February 2023, the FDA has approved 155 companion diagnostic devices for targeted drugs for patients with solid tumors and hematologic malignancies [79]. The presence of specific molecular aberrations is associated with FDA-approved drugs for breast cancer, cholangiocarcinoma, colorectal cancer, endometrial carcinoma, gastric and gastroesophageal cancer, head and neck cancer, medullary thyroid cancer, metastatic castrate-resistant prostate cancer, non-small cell lung cancer, ovarian cancer, pancreatic cancer, solid tumors, thyroid cancer, triple-negative breast cancer, urothelial cancer, and uveal melanoma. Similarly, the hematologic malignancies include acute myeloid leukemia, aggressive systemic mastocytosis, chronic myeloid leukemia, follicular lymphoma, and B-cell chronic lymphocytic leukemia. Examples of tumor types, FDA-approved drugs, and molecular alterations include the following: ovarian cancer, olaparib and rucabarib for BRCA1 and BRCA2 (blood); breast cancer, olaparib and talazoparib for BRCA1 and BRCA2 (blood); pancreatic cancer, olaparib for BRCA1 and BRCA2 (blood); NSCLC, adagrasib for KRAS G12C (plasma), osimertinib for EGFR T790M (tissue), osimertinib for EGFR exon 19 deletion or exon 21 L858R substitution mutation (tissue or plasma), and gefinitib for exon 19 deletion or exon 21 L858R substitution mutation; metastatic castrate-resistant prostate cancer, olaparib for BRCA1 and BRCA2 (blood); and melanoma, vemurafenib for BRAF V600E (tissue) and cobimetinib in combination with vemurafenib for BRAF V600E and BRAF V600K (tissue). The plethora of tumor types for which FDA-approved targeted drugs are available that inhibit the function of specific molecular aberrations provides access to matched targeted therapies to many patients with cancer [79]. Academic institutions and pharmaceutical companies continue to develop drugs and therapeutic strategies with innovative mechanisms of action to increase response rates and improve the PFS and OS in patients with cancer.

Despite the evidence that precision oncology is associated with superior outcomes in specific tumor types and diverse cancers, several gaps still exist. The challenges include the lack of universal use of molecular testing and modern technological advances to thoroughly understand the evolution of carcinogenesis in individual patients, and the lack of patient access to therapeutic strategies that would lead to the regression of this process and the elimination of cancer. Even though many therapies with biomarker selection are available (either FDA-approved or investigational through clinical trials), precision oncology is not accessible to all patients with cancer, and some patients’ tumors do not respond to these treatments. This lack of response can be attributed to the biological complexity of some tumors, which cannot be targeted with a single therapy, the absence of an effective targeted therapy, or an unknown mechanism of tumor resistance to treatment. Patient access to clinical trials can be limited by the prolonged turnaround time in receiving molecular testing results, a lack of insurance coverage, the expense of clinical trials, and the lack of drug availability. Implementing artificial intelligence, machine learning, and bioinformatic analyses of complex multi-omic data in clinical trials may improve the accuracy of the tumor characterization process. This will ultimately accelerate the implementation of precision medicine.

In conclusion, the results from completed and ongoing precision oncology trials, including the IMPACT/IMPACT2 studies, the randomized SHIVA study, NCI-MPACT, TAPUR, and NCI-MATCH, highlight the challenges and opportunities associated with the precision oncology approach. These efforts optimize the treatment options offered to patients, eliminating treatment selection bias and unlocking the full potential and value of precision oncology. As the field of precision oncology quickly evolves, the continuous assessment of the efficacy and toxicity of novel investigational agents targeting driver molecular alterations or critical mechanisms involved in carcinogenesis is needed. Some innovative therapies can induce deep responses with minimal or manageable toxicities. Clinical trials should be considered when the standard-of-care treatments have failed to confer clinical benefit or therapy has been discontinued owing to toxicity. Innovative trials, including randomized controlled studies that carefully consider the advantages and limitations of each design, may validate novel genomics-guided therapeutic strategies and accelerate the implementation of precision oncology. The clinical trials in precision oncology continue to evolve, improving the outcomes and expediting the identification of curative strategies for patients with cancer. Despite the existing challenges, significant progress has been made since the initiation of our precision oncology program, demonstrating the benefit of precision oncology in many patients with advanced cancer.

## Figures and Tables

**Figure 1 cancers-15-01967-f001:**
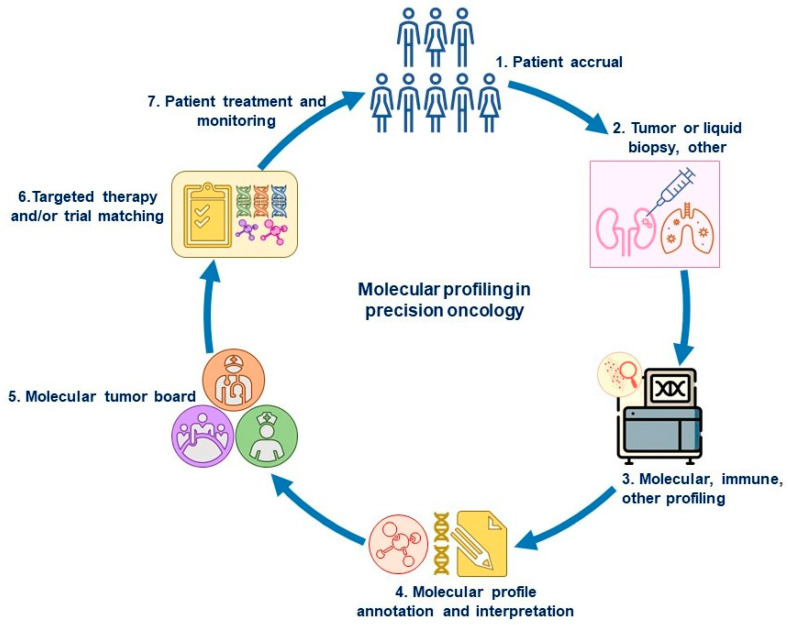
Overview of molecular profiling in precision oncology. In the screening phase of a biomarker-driven trial, patients (1) undergo a tumor biopsy or blood draw (liquid biopsy) (2) that is used for tumor molecular profiling (3) to determine the drivers of carcinogenesis (genomic or protein level), if any (4). The molecular profile report is often discussed at a molecular tumor board (5) for the interpretation of tumor alterations and for matching with a targeted therapy or clinical trial (6). The patient is then treated with the assigned therapy (FDA-approved or investigational drug after screening for clinical trial) and monitored for anti-tumor effects and toxicity (7). If the disease progresses, the next treatment can again be selected from a new round of tumor or blood analyses to identify evolving biomarkers.

**Figure 2 cancers-15-01967-f002:**
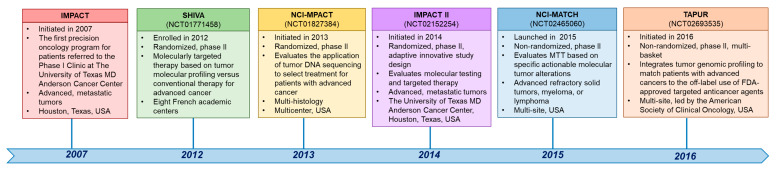
A timeline of clinical trials in precision oncology.

**Table 1 cancers-15-01967-t001:** Other selected trials in precision oncology across tumor types.

First, Last Author, Year	Treatment	Cancer Type	Molecular Alteration(s)	Enrollment (*N*)	Safety	Clinical Outcomes
**Pilot study using tumor molecular profiling (MP) to identify targets and matched treatments for refractory cancer**						
Von Hoff; Penny, 2010 [10]	Various treatments *	Diverse solid tumors	Oligonucleo-tide microarray (MA) gene expression assays	106	TRAE, n = 9 (anemia, n = 2; neutropenia, n = 2; dehydration, n = 1; pancreatitis, n = 1; nausea, n = 1; vomiting, n = 1; febrile neutropenia, n = 1) Treatment discontinuation, n = 1 (patient’s request, grade 2 fatigue).	Molecular target detected, 98% (84/86 had MP attempted); 66 of 84 patients were treated according to MP results; 18 (27%) of 66 patients had a PFS ratio (PFS on MP-selected therapy/PFS on prior therapy) of ≥1.3.
**IMPACT (Initiative for Molecular Profiling and Advanced Cancer Therapy)**						
Tsimberidou; Kurzrock, 2012 [9]	Matched versus unmatched therapy following NGS results	Diverse solid tumors	NGS	1144	N/A	ORR, matched vs. unmatched, 27% vs. 5% (*p* < 0.0001) mTTF, 5.2 vs. 2.2 months (*p* < 0.0001) mOS, 13.4 vs. 9.0 months (*p* = 0.017)
Tsimberidou; Berry, 2014 [11]	Matched versus unmatched therapy following NGS results	Diverse solid tumors	NGS	1542	N/A	ORR, matched vs unmatched, 12% vs. 5% (*p* < 0.0001) mPFS, 3.9 vs. 2.2 months (*p* = 0.001) mOS, 11.4 vs. 8.6 months (*p* = 0.04) Matched therapy independent factor for response and PFS mOS (2-month Landmark analysis): Responders vs. non-responders, 30.5 months vs 11.3 months (*p* = 0.01)
Tsimberidou; Kurzrock, 2017 [12]	Matched versus unmatched therapy following NGS results	Diverse solid tumors	NGS	1436	N/A	MTT (n = 390), non-MTT (n = 247) MTT vs. non-MTT, ORR (PR): 11% vs. 5% (*p* = 0.0099) FFS, 3.4 vs. 2.9 months (*p* = 0.0015) OS, 8.4 vs. 7.3 months (*p* = 0.041)
Tsimberidou; Kurzrock, 2019 [13]	Matched versus unmatched therapy following NGS results	Diverse solid tumors	NGS	3487	N/A	ORR, MTT 16.4%, non-MTT 5.4% (*p* < 0.0001) SD ≥ 6 months, MTT 35.3%, non-MTT 20.3%, (*p* < 0.001) mPFS, MTT 4.0, non-MTT 2.8 months (*p* < 0.0001) MTT vs. non-MTT: mOS, 9.3 months vs. 7.3 months; 3-yr OS, 15% vs. 7%; 10-yr OS, 6% vs. 1% (*p* < 0.0001)
**Investigation of Profile-Related Evidence Determining Individualized Cancer Therapy (I-PREDICT)**						
Schwaederle; Kurzrock, 2016 [14]	Matched versus unmatched therapy following NGS results	Metastatic/ refractory, therapy-naive solid tumors	PD-L1, ctDNA, TMB, MSI	347	N/A	DCR, matched, 34.5% vs. unmatched therapy, 16.1% (*p* ≤ 0.005); mPFS, months: matched 4.0 vs. unmatched therapy, 3.0 (*p* = 0.039); mOS, months: matched, 15.7 (matching score > 0.2) vs. unmatched, 10.6 months (matching-score of ≤ 0.2) (*p* = 0.040)
**Worldwide Innovative Network trial with genomics and transcriptomics (WINTHER)**						
Rodon; Kurzrock, 2019 [15]	Matched versus unmatched therapy following MP results	Diverse advanced metastatic solid tumors	NGS and transcriptomics	303	High-grade AEs: diarrhea, rash, fatigue/weakness	Evaluable for treatment, n = 107 (35%; arm A, n = 69; arm B, n = 38). SD ≥ 6 months/PR/CR, 26.2% (arm A: 23.2%; arm B: 31.6% (*p* = 0.37)). PFS ratio (patient proportion with WINTHER vs. previous therapy of >1.5) = 22.4%
**Dual Anti-CTLA-4 and Anti-PD-1 blockade in Rare Tumors (DART)**						
Patel; Kurzrock, 2020 [16]	Ipilimumab + nivolumab	Non-pancreatic neuro-endocrine carcinoma	Dual anti-CTLA-4 and anti-PD-1	32 Most common primary sites: GI (47%; N = 15); lung (19%; N = 6)	G3/4 AEs: hypothyroidism (31%), fatigue (28%), ALT elevation (9%)	SD, 41% (6%, SD ≥ 6 months) ORR, 25% (CR, 3%, n = 1; PR, 22%, n = 7) High-grade: ORR, 44% Low/intermediate-grade: ORR, 0% 6-month PFS: 31% Evaluable for OS, N = 18 Median OS: 11 months (95 CI, 6-NR)
**Octopus study, phase I/II**						
M. Wrangle; Soon-Shiong, 2021 [17]	Quilt-3.055 (NCT03228667) N803 (IL-15 superagonist) plus investigator choice †	Diverse solid tumors	T-cell modulation, PD-L1	135 (60% NSCLC)	G1-2 TRAE: injection site reaction 68%, chills 32%, fatigue 26%, pyrexia 26%, flu-like illness 14%, nausea 12%	Response: non-evaluable, 12% CR 0%, PR 8%, SD 51%, PD 29% PFS, median, 3.9 months OS, median, 13.8 months

Abbreviations: AE, adverse event; ALT, alanine aminotransferase; ALK, receptor tyrosine kinase; CR, complete response; DCR, disease-control rate; EGFR, encoding epidermal growth factor receptor; FFS, failure-free survival; GI, gastrointestinal; G, grade; mOS, median overall survival; MP, molecular profiling; MSI, microsatellite instability; mPFS, median progression-free survival; mTTF, median time-to-treatment failure; MTT, matched targeted therapy; NAB, nanoparticle albumin-bound; N-803, IL-15 superagonist; NGS, next-generation sequencing; OS, overall survival; OR, objective response; ORR, objective response rate; PD-L1, programmed cell death ligand 1; PFS, progression-free survival; PR, partial response; SD, stable disease; TMB, tumor mutational burden; TRAE, treament-related adverse event; t-haNK, PD-L1 targeting high-affinity NK; TTF, time-to-treatment failure; * diethylstilbestrol, NAB paclitaxel + trastuzumab, NAB paclitaxel + gemcitabine, letrozole + capecitabine, oxaliplatin + fluorouracil + trastuzumab, gemcitabine + pemetrexed, doxorubicin, exemestane, irinotecan + sorafenib, temozolomide + bevacizumab, sunitinib + mitomycin, temozolomide + sorafenib, lapatinib + tamoxifen, cetuximab + irinotecan, cetuximab + irinotecan, gemcitabine + etoposide, sunitinib, cetuximab + gemcitabine; † N-803 + pembrolizumab/nivolumab/atezolizumab/avelumab/durvalumab/pembrolizumab + PD-L1 thaNK/nivolumab + PD-L1 t-haNK/atezolizumab + PD-L1 t-haNK/avelumab + PD-L1 t-haNK/durvalumab + PD-L1 t-haNK.

**Table 2 cancers-15-01967-t002:** Subprotocols of the non-randomized NCI-MATCH and TAPUR studies across tumor types with positive results.

Published Data (First, Last Author, Year)	Treatment	Cancer Type	Molecular Alteration(s)	Enrollment (*N*)	Safety	Clinical Outcomes
**NCI-MATCH** (as of 13 August 2022 update, 39 arms: 2 open, 12 published, 10 presented, 15 closed)						
Azad; Flaherty, 2020 [35]	Nivolumab	Non-CRCs	Mismatch repair-deficient	42	G4 toxicities, n = 3 (sepsis, n = 2) G1-3 AEs: fatigue (40%), anemia (33%), rash (17%), hypoalbuminemia (17%)	ORR, 36%; SD, 21%; 6-month PFS, 51.3%; 12-month PFS, 46.2%; 18-month PFS, 31.4%; Median OS, 17.3 months
Kalinsky; Flaherty, 2021 [41]	Capivasertib	Diverse tumor types	AKT1 E17K mutations	35	Discontinued: AEs, 31% (11/35); G3 treatment-related AE: hyperglycemia (n = 8, 23%) and rash (n = 4, 11%); G4 hyperglycemia (n = 1)	ORR, 28.6%; 6-month PFS, 50%; Median OS, 14.5 months
Salama; Flaherty, 2020 [42]	Dabrafenib and trametinib	Diverse tumor types	BRAF V600E mutations	35	G3 AEs: fatigue (n = 4); neutropenia (n = 3); hyponatremia (n = 2); G4 sepsis (n = 1).	ORR, 38%; Median PFS, 11.4 months; Median OS, 28.6 months
Damodaran; Flaherty, 2022 [43]	Copanlisib	Diverse tumor types	PIK3CA mutations	35 (25 were included in the primary efficacy analysis as prespecified in the protocol)	G3 AEs: hyperglycemia (n = 7); rash, maculopapular (n = 2); mucositis, oral (n = 1); vomiting (n = 1); weight loss (n = 1); general muscle weakness (n = 2); pruritus (n = 1); hypertension (n = 9); dehydration (n = 2); acute kidney injury (n = 1); dizziness (n = 1); hypophosphatemia (n = 1); hypoglycemia (n = 1); hypoxia (n = 1); meningitis (n = 1); oral pain (n = 1); syncope (n = 1). G4 AE: hyperglycemia (n = 1)	ORR, 16%; Median PFS, 3.4 months; median OS, 5.9 months
Mansfield; Flaherty, 2022 [45]	Crizotinib	Diverse tumor types	ALK or ROS1 rearrangements	9 (5 ALK, 4 ROS1)	ALK: G3 AEs: AST increased (n = 1); hypophosphatemia (n = 1) G4 AEs: ALT increased (n = 1); AST increased (n = 1); blood bilirubin increased (n = 1); hyponatremia (n = 1) ROS1: G3 AEs: abdominal pain (n = 1); ALC decreased (n = 1); acute kidney injury (n = 1)	ALK: ORR, 50%; median PFS, 3.8 months; median OS, 4.3 months ROS1: ORR, 25%; median PFS, 4.3 months; median OS, 6.2 months
**TAPUR** **https://www.asco.org/research-data/tapur-study/study-results, accessed on 9 August 2022, cohort updated 8 April 2022**						
Pisick; Schilsky, Meeting Abstract, 2020 [51]	Olaparib	Prostate cancer	BRCA1/2 inactivating mutations	29 (25 included in efficacy analyses)	G3-4 AE (n = 6): anemia, aspiration, dehydration, diabetic ketoacidosis, fatigue, and neutropenia	DCR, 68%; ORR, 36%; median PFS, 41.0 weeks; median OS, 75.4 weeks; 1-year OS rate, 79.4%
Ahn; Schilsky, Meeting Abstract, 2020 [52]	Olaparib	Pancreatic cancer	BRCA1/2 inactivating mutations	30 (28 included in efficacy analyses)	G3-4 AE (n = 7): anemia, diarrhea, fever, elevated liver enzymes, enterocolitis, increased bilirubin, oral mucositis	DCR, 31%; ORR, 4%; median PFS, 8.1 weeks; median OS, 43 weeks; 1-year OS rate, 47.2%
Mileham; Schilsky Meeting Abstract, 2022 [53]	Olaparib	Diverse tumor types	ATM mutations or deletions	39 (37 included in efficacy analyses)	G3-4 AE or SAE (n = 9): anemia, anorexia, colitis, dehydration, dizziness, fatigue, hypokalemia, lung infection, nausea, proteinuria, urinary tract infection, urinary tract obstruction	DCR, 27%; ORR, 8%; median PFS, 8.6 weeks; median OS, 40.9 weeks
Ahn; Schilsky, Meeting Abstract, 2022 [54]	Olaparib	Diverse tumor types	Germline or somatic BRCA1/BRCA2 inactivating mutations	32 (32 included in efficacy analyses)	G3-4 AE or SAE (n = 12): anemia, dyspnea, fatigue, fever, generalized muscle weakness, tumor lysis syndrome, leukopenia/thrombocytopenia	DCR, 41%; ORR, 25%; median PFS, 15.7 weeks; median OS, 45 weeks
Pisick; Schilsky, Meeting Abstract, 2021 [55]	Palbociclib	Head and neck cancer	CDKN2A loss or mutation	28 (28 included in efficacy analyses)	G3-4 AEs (n = 13): Cytopenias, hypocalcemia, syncope G5 AEs (n = 1): respiratory failure	ORR, 0%; DCR, 37%; median PFS, 9.4 weeks; median OS, 42.0 weeks
Ahn; Schilsky, 2020 [56]	Palbociclib	Non-small cell lung cancer	CDKN2A alterations	29 (27 included in efficacy analyses)	G3-4 AE or SAE (n = 11): most common, cytopenias	DCR, 31%; median PFS, 8.1 weeks; median OS, 21.6 weeks
Schuetze; Schilsky, Meeting Abstract, 2021 [57]	Palbociclib	Soft tissue sarcoma	CDK4 amplification	29 (28 included in efficacy analyses)	G3-4 AEs (n = 14): most common, leukopenia/thrombocytopenia	DCR, 48%; ORR, 3.7%; median PFS, 16.1 weeks; median OS, 68.7 weeks; 1-year OS rate, 53.6%
Gupta; Schilsky, Meeting Abstract, 2020 [58]	Pertuzumab and trastuzumab	CRC	ERBB2 amplification or overexpression	28 (28 included in efficacy analyses)	G3 SAE: anemia, infusion reaction, left ventricular dysfunction	DCR, 50%; ORR, 14%; median PFS, 17.2 weeks; 1-year OS rate, 58%
Ali-Ahmad; Schilsky, Meeting Abstract, 2021 [59]	Pertuzumab and trastuzumab	Uterine cancer	ERBB2 or ERBB3 amplification, overexpression, or mutation	28 (28 included in efficacy analyses)	G3 AE (n = 1): muscle weakness	DCR, 37%; ORR, 7.1%; median OS, 28.1 weeks; 1-year OS rate, 53.4%
Gant; Schilsky, Meeting Abstract, 2022 [60]	Pertuzumab and trastuzumab	Bronchus and lung	ERBB2/ERBB3 amplification, mutation, or overexpression	28 (28 included in efficacy analyses)	G3-4 AE or SAE (n = 5): ALT increased, AST increased, dyspnea, fatigue, infusion-related reaction, nausea, vomiting	DCR, 37%; ORR, 11%; Median OS, 54.4 weeks
Alva; Schilsky, 2021 [61]	Pembrolizumab	Metastatic breast cancer	High tumor mutational burden	28 (28 included in efficacy analyses)	G3 (n = 4): pulmonary embolism, weight loss, hypoalbuminemia, hyponatremia G2-3 SAE (n = 4): colonic obstruction, diarrhea, urinary tract infection, hepatic failure	DCR, 37%; ORR, 21%; median PFS,10.6 weeks; median OS, 30.6 weeks
Meiri; Schilsky, Meeting Abstract, 2020 [62]	Pembrolizumab	CRC	High tumor mutational burden	28 (27 included in efficacy analyses)	G3 AE (n = 2, each): abdominal infection, anorexia, colitis, diarrhea, fatigue, nausea, vomiting G3 SAE (n = 1): acute kidney injury	DCR, 28%; ORR, 4%; median PFS, 9.3 weeks; 1-year OS rate, 45.6%
Calfa; Schilsky, Meeting Abstract, 2021 [63]	Sunitinib	Metastatic breast cancer	FGFR1 alterations	30 (27 included in efficacy analyses)	G3 AEs (n = 9): cytopenia, encephalopathy, febrile neutropenia, increased alkaline phosphatase, palmar-plantar erythrodysesthesia syndrome, vomiting G4 AEs (n = 2): cytopenia, hypertension	DCR, 29%; ORR, 7%; median PFS, 8.7 weeks; median OS, 33.9 weeks
Klute; Schilsky, Meeting Abstract, 2020 [64]	Cobimetinib and vemurafenib	CRC	BRAF V600E mutations	30 (28 included in efficacy analyses)	G3 AE/SAE (n = 12): elevated liver enzymes, decreased lymphocytes, dyspnea, diarrhea, fatigue, hypercalcemia, hypophosphatemia, rash, photosensitivity, upper GI hemorrhage, vomiting	DCR, 57%; ORR, 29%; median PFS, 15.8 weeks; median OS, 38.9 weeks
Meric-Bernstam; Schilsky, Meeting Abstract, 2022 [65]	Cobimetinib and vemurafenib	Diverse tumor types	BRAF_V600E/D/K/R mutation	31 (28 included in efficacy analyses)	G3 AE (n = 17): rash, anemia, hypokalemia, increased ALP, increased AST, increased ALT, increased CPK, diarrhea, increased GGT, hypophosphatemia, decreased ALC, multiple SCCs of skin, decreased platelet count, treatment-related secondary malignancy G4 AE (n = 1): increased GGT	DCR, 68%; ORR, 57%; median PFS, 5.8 months; median OS, 15.2 months

Abbreviations: AE, adverse event; AKT1, AKT serine/threonine kinase 1; ALC, absolute lymphocyte count; ALK, ALK receptor tyrosine kinase; ALP, alkaline phosphate; ALT, alanine aminotransferase; AST, aspartate aminotransferase; ATM, Ataxia-telangiectasia mutated; BC, breast cancer; BRCA1/2, breast cancer gene 1/2; BRAF, B-Raf proto-oncogene, serine/threonine kinase; CDK4, cyclin dependent kinase; CDKN2A, cyclin dependent kinase inhibitor 2A; CPK, creatine phosphokinase; CRC, colorectal cancer; DCR, disease control rate; ERBB2 (HER2), Erb-B2 receptor tyrosine kinase 2; FGFR, fibroblast growth factor receptor; FLT-3, Fms-related receptor tyrosine kinase 3; G, grade; GI, Gastrointestinal, GGT, gamma glytamyl-transferase; KRAS, KRAS proto-oncogene, GTPase; mTOR, mechanistic target of rapamycin kinase; Non-CRC, non-colorectal cancer; ORR, objective response rate; OS, overall survival; PFS, progression-free survival; PICK3CA, phosphatidylinositol-4,5-bisphosphate 3-kinase catalytic subunit alpha; ROS1, ROS proto-oncogene 1, receptor tyrosine kinase.

**Table 3 cancers-15-01967-t003:** Selected subprotocols of the non-randomized NCI-MATCH and TAPUR studies across tumor types with negative results.

NCI-MATCH						
**First, Last Author, Year**	**Treatment**	**Cancer Type**	**Molecular Alteration(s)**	**Enrollment (*N*)**	**Safety**	**Clinical Outcomes**
Chae; Flaherty, 2020 [47]	AZD4547 (FGFR inhibitor)	Diverse tumor types	FGFR pathway aberrations	70 (48 eligible and treated)	G3 AEs: oral mucositis (n = 7); constipation (n = 1); dry eye (n = 1); anemia (n = 2); palmar-plantar erythrodysesthesia (n = 3); peripheral sensory neuropathy (n = 1); dizziness (n = 1); abdominal pain (n = 1); esophageal pain (n = 1); small intestinal obstruction (n = 1); laryngeal mucositis (n = 1); syncope (n = 1); febrile neutropenia (n = 1); increased ALP (n = 1); increased ALT (n = 3); increased AST (n = 4); hypernatremia (n = 1); hypophosphatemia (n = 1); decreased neutrophil count (n = 1); increased GGT (n = 1) G4 AEs: diarrhea (n = 1), sepsis (n = 1)	6-month PFS, 15%; median PFS, 3.4 months
Johnson; Flaherty, 2020 [46]	Trametinib	Solid tumors and lymphomas	BRAF non-V600 mutations or fusions	50 (32 eligible and treated)	G3 AEs: anemia (n = 5); nausea (n = 1); vomiting (n = 1); anorexia (n = 1); hypoalbuminemia (n = 1); pruritus (n = 1); rash acneiform (n = 1); rash maculopapular (n = 1); other skin and subcutaneous tissue disorders (n = 1)	6-month PFS, 17%; median PFS, 1.8 months; 6-month OS, 46%; median OS, 5.7 months
Jhaveri; Flaherty, 2019 [37]	Ado-trastuzu-mab emtansine (T-DM1)	Diverse tumor types other than breast and gastroeso-phageal tumors	HER2 amplification at a copy number >7	38 (36 included in efficacy analysis)	G3 AEs: anemia (n = 3); fatigue (n = 2); fever (n = 1); nausea (n = 1); ileal obstruction (n = 1); ALP increase (n = 1); AST increase (n = 1); lymphocyte count decrease (n = 1); neutrophil count decrease (n = 1); platelet count decrease (n = 2); anorexia (n = 2); epistaxis (n = 1); hypoxia (n = 1); muscle weakness, lower limb (n = 1); dehydration (n = 1); investigations, other (n = 1); urinary tract infection (n = 1); upper respiratory infection (n = 1); diarrhea (n = 1); blurred vision (n = 1)	6-month PFS, 23.3%; median OS, 8.4 months; ORR, 5.6%
Bedard; Flaherty, 2022 [50]	Afatinib	Diverse tumor types	ERBB2-activating mutations	59 (40 enrolled, 37 included in efficacy analysis)	G3 AEs: diarrhea (18.9%), mucositis (8.1%), and fatigue (8.1%)	6-month PFS, 12.0%; median PFS, 1.7 months; median OS, 6.5 months; ORR, 2.7%
Cleary; Flaherty, 2021 [48]	Binimetinib	Diverse tumor types (melanoma excluded)	Codon 12/13 and codon 61 NRAS-mutated	53 (47 eligible and included in efficacy analysis)	G3 AEs: heart failure (n = 1); myocardial infarction (n = 1); eye disorders (n = 1); mucositis oral (n = 1); nausea (n = 1); small intestinal obstruction (n = 1); fatigue (n = 1); edema limbs (n = 1); urinary tract infection (n = 1); ALT increased (n = 1); ALP increased (n = 1); AST increased (n = 1); CPK increased (n = 2); lymphocyte count decreased (n = 2); white blood cell decreased (n = 1); ejection fraction decreased (n = 1); anorexia (n = 1); dehydration (n = 1); hypoalbuminemia (n = 1); hyponatremia (n = 1); hypophosphatemia (n = 1); muscle weakness, lower limb (n = 1); muscle weakness, upper limb (n = 1); syncope (n = 1); rash acneiform (n = 1); skin and subcutaneous tissue disorders (n = 1); hypertension (n = 6) G5 AE: multi-organ failure (n = 1)	6-month PFS, 29.2%; median PFS, 3.5 months; median OS, 10.5 months; ORR, 2.1%
Krop; Flaherty, 2022 [49]	Taselisib	Solid tumors other than breast and squamous cell lung cancer	PIK3CA mutations	70 (61 eligible and initiated protocol)	G3 AEs: diarrhea (n = 7); fatigue (n = 1); nausea (n = 2); hyperglycemia (n = 2); anorexia (n = 1); mucositis, oral (n = 1); AST increased (n = 1); abdominal pain (n = 1); vomiting (n = 1); hypokalemia (n = 1); hyponatremia (n = 3); dehydration (n = 2); hypertension (n = 1); weight loss (n = 1); lung infection (n = 3); pneumonitis (n = 1); thromboembolic event (n = 1); adult respiratory distress syndrome (n = 1); blood bilirubin increased (n = 1); dysphagia (n = 1) G4 AEs: hyperglycemia (n = 1) G5 AEs: neoplasms benign, malignant, and unspecified (n = 1); sudden death NOS (n = 1)	6-month PFS, 19.9%; median PFS, 3.1 months; 6-month OS, 60.7%; median OS, 7.2 months
TAPUR						
Baghdadi; Schilsky 2019 [66]	Palbociclib	Pancreatic	CDKN2A loss or mutation	12 (10 evaluable patients)	≥G3 AEs (n = 1): fatigue	No patients had objective response or stable disease at 16 weeks. Median PFS, 7.2 weeks; median OS, 12.4 weeks
Baghdadi; Schilsky 2019 [66]	Palbociclib	Biliary cancers	CDKN2A loss or mutation	10 (10 evaluable patients)	G3 (n = 1): muscle weakness and port infection; G4 (n = 4): thrombocytopenia	No patients had objective response or stable disease at 16 weeks. Median PFS, 7.3 weeks; median OS, 11.1 weeks
Baghdadi; Schilsky 2020 [67]	Sunitinib	Metastatic CRC	FLT-3 amplification	10 (10 evaluable patients)	G3 AEs (n = 1): diarrhea	Median PFS, 10.1 weeks; median OS, 38 weeks
Fisher; Schilsky, 2020 [68]	Cetuximab	Advanced breast, NSCLC, and ovarian cancer	KRAS, NRAS, BRAF mutations	49 (28 evaluable patients)	≥G3 AEs (n = 6) BC: hypomagnesemia; NSCLC: anemia, hyponatremia, hypophosphatemia, hypokalemia, hypomagnesemia, and cytopenia; OC: fever, infusion-related reaction, hypotension, nausea, vomiting	BC: Median PFS, 6.7 weeks; median OS, 14.1 weeks NSCLC: Median PFS, 8 weeks; median OS, 22.7 weeks OC: Median PFS, 8 weeks; median OS, 21.6 weeks
Vaccaro; Schilsky, Meeting Abstract, 2019 [69]	Nivolumab and ipilimumab	CRC	High tumor mutational burden	12 (10 included in efficacy analyses)	G3-4 AEs (n = 4): myasthenia gravis, diarrhea, glucose intolerance, hyperglycemia, small intestinal obstruction	DCR, 10%; ORR, 10%; median PFS, 8.9 weeks; median OS, 42.9 weeks
Grem; Schilsky, Meeting Abstract, 2022 [70]	Temsiro-limus	CRC	PIK3CA mutation	10 (10 included in efficacy analyses)	G3-4 AEs (n = 6): acute kidney injury, dehydration, thrombocytopenia, hypertriglyceridemia, mucositis, neutropenia, scrotal and penile edema	DCR, 10%; ORR, 0%; median PFS, 8.1 weeks; median OS, 38.7 weeks
Srkalovic; Schilsky, Meeting Abstract, 2022 [71]	Temsiro-limus	Diverse cancer types	mTOR mutation/ amplification	29 (20 included in efficacy analyses)	G3-4 AEs (n = 8): acute kidney injury, epistaxis, hyperglycemia, hypertension, hypertriglyceridemia, oral mucositis, leukopenia, thrombocytopenia, pneumonitis	DCR, 45%; ORR, 10%; median PFS, 16.1 weeks; median OS, 48.7 weeks

Abbreviations: AKT1, AKT serine/threonine kinase 1; ALC, absolute lymphocyte count; ALK, ALK receptor tyrosine kinase; ALP, alkaline phosphate; ALT, alanine aminotransferase; AST, aspartate aminotransferase; BC, breast cancer; BRAF, B-Raf proto-oncogene, serine/threonine kinase; CDKN2A, cyclin dependent kinase inhibitor 2A; CPK, creatine phosphokinase; CRC, colorectal cancer; DCR, disease control rate; ERBB2 (HER2), Erb-B2 receptor tyrosine kinase 2; FGFR, fibroblast growth factor receptor; FLT-3, Fms-related receptor tyrosine kinase 3; G, grade; GGT, gamma glytamyl-transferase; KRAS, KRAS proto-oncogene, GTPase; mTOR, mechanistic target of rapamycin kinase; NRAS, NRAS proto-oncogene, GTPase; OC, ovarian cancer; ORR, objective response rate; OS, overall survival; PFS, progression-free survival; PICK3CA, phosphatidylinositol-4,5-bisphosphate 3-kinase catalytic subunit alpha; ROS1, ROS proto-oncogene 1, receptor tyrosine kinase.

## Data Availability

The data presented in this study are available in this article.

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
