# Peer review of "Precision Oncology: Evolving Clinical Trials across Tumor Types"

_cancers, 2023, doi:10.3390/cancers15071967_

Round 1

Reviewer 1 Report

Song and colleagues have assembled a review article focusing on the evolution of precision oncology clinical trials in the field of personalized medicine. This is a nice subject as several trials have run to completion and there are more patients benefiting from the findings (and participation in) of these clinical trials. Overall, I believe there needs to be much more detail to establish to the reader how the trials evolved, as proposed in the article title, and where the gains and losses/lessons learned were found in the trials reviewed. I detail a few examples below. If this can be addressed, this manuscript will service the scientific community well and I commend the authors for proposing it.

Section 2:

The content here is important, but it lacks specificity. Aspects of what motivates these kinds of trials and fully describing how each of the challenges listed in this section require at least one detailed example to illustrate the concepts. Related to this, there is no description or comparison to pre-precision oncology clinical trials, nor defining what a biomarker is in the context of cancer and how they are used (diagnostic or treatment applications). Finally, related to Figure 1, the authors should provide more written details about each step – the reader is only left with the figure. If all of the above is addressed, then the subsequent sections in the review are easier to comprehend.

Section 3:

·      Section 3.1 needs significantly more background for its motivation and details of what cancers and mutations the study was looking at. It also needs to elaborate on the patient population being studied, specific criteria for inclusion and exclusion, as well as much more detailed summarization of the results and conclusions drawn from the study (e.g. treatments pursued, how it integrated into clinical work flow). If this was the first-ever precision oncology trial, it is important to highlight how it influenced future studies and where a lot of breakdown came from prior subsequent studies presumably containing iterative improvements.

·      Sections 3.2 through 3.6 are a little more detailed than 3.1, but again, if this review paper is showing how precision oncology trials evolved, there needs to be in-depth descriptions about all aspects of the study, their successes and the lessons learned from them. The reader, with this draft, gets a few genes of interest and a fair amount of statistical findings but there is no contextualization in any of the paragraphs (albeit there are few good examples within the existing text).

Section 4:

I would recommend more detail be provided about these listed trials. Again, link to lessons learned from the past and summarize what specifically is being looked at, treatment alternatives, etc.

Section 5:

I think there is tremendous opportunity to be had with this section. For example, the molecular diagnostic workflow, machines, algorithms are assumed to be different between trials and treatment centers. Go into specifics about how these differences influence the findings, what is being done to bring standardization to the precision oncology world. Additionally, after all of these trials, what has happened with regards to patient access (you make a reference statement, but no examples or suggestions for improvement)? How has the FDA changed in time with these trials? What is pharma doing now that this data is out? These are just a few examples, but more thought is needed into developing this section.

General statement:

There is a lot of “we” in this paper – this should be changed unless the authors were not running these trials.

Reviewer 2 Report

Dear Authors, 

I read with pleasure your review on clinical trials in precision oncology, that is well written and complete. I would only suggest to add a little paragraph on the DRUP protocol, whose personalized reimbursement model I think deserves to be mentioned when talking about precision oncology trials' challenges and opportunities.

Best regards

Reviewer 3 Report

In this manuscript, the Authors aimed to summarize some selected precision oncology clinical trials (including the IMPACT, 15 SHIVA, IMPACT2, NCI-MPACT, TAPUR, and NCI-MATCH studies), discussing the challenges and opportunities of this field.

Although the Authors reported the results of the above-mentioned clinical trials in detail, this review has a significant selection bias. Indeed, why have the Authors reported only the results of the above-mentioned clinical trials? Despite the “narrative” and not “systematic” nature of this review, why have other published precision oncology trials not been reported in this manuscript? In this scenario, I also suggest removing the “our” from “our IMPACT studies”.

In addition, I suggest the Authors add a brief “Discussion” paragraph where they can speculate and discuss the results of the studies included in this review.

Furthermore, I suggest adding a title for the subparagraph related to the SHIVA trial. All the above-mentioned trials had their own subparagraph title apart from the SHIVA trial.

Finally, the references in the reference list are not reported correctly. Based on the guidelines of this journal, all the Authors’ names must be explicated in the references. Therefore, it is difficult to check for the presence of auto-citations. In this direction, 6 out of 74 references (number 8-9-10-11-12-14) are authored by the last co-author of this manuscript.

Round 2

Reviewer 1 Report

The reviewer greatly appreciates the effort put forth by the authors to address the first round review comments. I am satisfied with the results and approve the manuscript for publication.

Reviewer 3 Report

The Authors have addressed adequately my concerns.